# Systematic discovery of conservation states for single-nucleotide annotation of the human genome

Adriana Arneson[1,2] & Jason Ernst [1,2,3,4,5,6]

Comparative genomics sequence data is an important source of information for interpreting genomes. Genome-wide annotations based on this data have largely focused on univariate scores or binary elements of evolutionary constraint. Here we present a complementary whole genome annotation approach, ConsHMM, which applies a multivariate hidden Markov model to learn de novo 'conservation states' based on the combinatorial and spatial patterns of which species align to and match a reference genome in a multiple species DNA sequence alignment. We applied ConsHMM to a 100-way vertebrate sequence alignment to annotate the human genome at single nucleotide resolution into 100 conservation states. These states have distinct enrichments for other genomic information including gene annotations, chromatin states, repeat families, and bases prioritized by various variant prioritization scores. Constrained elements have distinct heritability partitioning enrichments depending on their conservation state assignment. ConsHMM conservation states are a resource for analyzing genomes and genetic variants.

[1] Bioinformatics Interdepartmental Program, University of California, Los Angeles, Los Angeles, CA 90095, USA. [2] Department of Biological Chemistry, University of California, Los Angeles, Los Angeles, CA 90095, USA. [3] Eli and Edythe Broad Center of Regenerative Medicine and Stem Cell Research at University of California, Los Angeles, Los Angeles, CA 90095, USA. [4] Computer Science Department, University of California, Los Angeles, Los Angeles, CA 90095, USA. [5] Jonsson Comprehensive Cancer Center, University of California, Los Angeles, Los Angeles, CA 90095, USA. [6] Molecular Biology Institute, University of California, Los Angeles, Los Angeles, CA 90095, USA. Correspondence and requests for materials should be addressed to J.E. (email: jason.ernst@ucla.edu)

The large majority of phenotype-associated variants implicated by genome-wide association studies (GWAS) are non-coding[1]. Identifying and interpreting causal non-coding variants is an important challenge[2]. Mapping of epigenomic data across different cell and tissue types has been one approach for annotating and interpreting the non-coding regions of genomes[3–5]. Using comparative genomics data to identify regions of evolutionary constraint has been a complementary approach for these purposes[6–9].

In addition to providing evolutionary information, comparative genomics data has the advantage of providing information at single-nucleotide resolution. Furthermore, it is cell type agnostic and thus informative even when the relevant cell or tissue type has not been experimentally profiled[10,11]. The most commonly used representations of this information are univariate scores and binary elements of evolutionary constraint, which are called based on a multiple species DNA sequence alignment and assumed models of evolution and selection[8,9,12–14]. Supporting the importance of these annotations, heritability analyses have recently implicated evolutionary constrained elements as one of the annotations most enriched for phenotype-associated variants[15]. These scores and elements have also been highly informative features to integrative methods for prioritizing pathogenic variants[16–19]. Further improvements to pathogenic coding variant prioritization scores have been made by also using features defined directly from a multiple sequence alignment[20].

While useful, the representation of comparative genomics information into univariate scores or binary elements is limited in the amount of information it can convey about the underlying multiple sequence alignment at a specific base. This limitation has become more pronounced given the large number of species now available in multi-species alignments such as a 100-way alignment to the human genome[21]. Approaches have been developed to associate constrained elements, regions, or individual bases with specific branches in a phylogenetic tree[22–28]. While also useful, such directed approaches are biased to only representing certain types of patterns present in an alignment. An alternative approach learned patterns of different classes of mutations between human and only one non-human genome[29], and was only applicable at a broad region level.

Analogous to the many sequenced genomes available for comparative analysis, many different epigenomic datasets are available for annotating genomes. Approaches that define 'chromatin states' based on combinatorial and spatial patterns in these datasets have effectively summarized the information in them to provide de novo genome annotations[4,30–32]. Inspired by the success of these approaches, here we develop a method, ConsHMM, that extends the ChromHMM[31] method to systematically annotate genomes into 'conservation states' at single nucleotide resolution given a multiple species DNA sequence alignment. ConsHMM takes a relatively unbiased and flexible modeling approach that does not explicitly assume a specific phylogenetic relationship between species.

We applied ConsHMM to assign a conservation state to each nucleotide of the human genome. The states capture distinct enrichments for other genomic annotations such as gene annotations, CpG islands, repeat families, chromatin states, genetic variation, and bases prioritized by variant prioritization scores. The ConsHMM conservation state annotations are a resource for interpreting genomes and potential disease-associated variation, which complement both existing conservation and epigenomic-based annotations.

## Results

### Annotating the human genome into conservation states. We developed an approach, ConsHMM, to annotate a genome into

conservation states at single nucleotide resolution based on a multiple species DNA sequence alignment (Fig. 1a, Methods). At each position in a reference genome, ConsHMM encodes one of three observations for each non-reference species in the alignment: aligns with a nucleotide present that is the same as the reference genome, different than the reference genome, or does not have a nucleotide present at that position. ConsHMM then probabilistically models the combinatorial and spatial patterns in these observations using a multivariate hidden Markov model (HMM). In each state of the HMM, ConsHMM assumes that the probability of observing a specific combination of observations is determined by a product of independent multinomial random variables. The parameter values will generally differ between states, and ConsHMM learns them from the input. After the model is learned, ConsHMM assigns each nucleotide in the reference genome to the state that had the maximum posterior probability of generating the observations.

We applied ConsHMM to a 100-way Multiz vertebrate alignment with the human genome as the reference genome[21,33]. We focused our analysis here on a model learned using 100 states to balance recovery of additional biological features and model tractability (Fig. 2, Supplementary Figs. 1–8, Methods). We verified that ConsHMM's transition parameters have a smoothing effect, which is consistent with applications of HMMs for constrained element detection[9,14], as the number of segments increased from 889 million to 1.06 billion when using an equivalent model except without transition information, though most state assignments to individual bases were the same (Supplementary Fig. 9, Methods). We illustrate ConsHMM conservation state annotations at two loci, which shows that bases with similar existing constraint annotations can have different conservation state assignments corresponding to very different underlying alignment patterns (Fig. 1b, Supplementary Fig. 10).

**Major groups of conservation states**. Hierarchically clustering the conservation states revealed eight notable subsets of states (Fig. 2a, Supplementary Fig. 4, Supplementary Data 1, Methods). The first subset was a single state (state 1) that showed high align and match probabilities through essentially all the vertebrates. The second subset showed relatively high align and match probabilities for all mammals and some non-mammalian vertebrates (states 2–4). The third subset showed relatively high align and match probabilities for most if not all mammals, but not non-mammalian vertebrates (states 5–22). The fourth subset showed high align probabilities for many mammalian species, but had low align probabilities for notable mammals such as mouse and rat for many of the states in the group (states 23–46). The lower mouse and rat probabilities relative to mammals that diverged earlier is consistent with increased substitution rates for mouse and rat[7]. The fifth subset showed high align probabilities for many mammalian species, but did not show high match probabilities (states 47–63). The sixth subset showed high align probabilities for most primates, but not for other species (states 64–89). The seventh subset showed high align probabilities for at most a subset of primates (states 90–99). The final subset was a single state (state 100) that showed high align and match probabilities for most primates and non-mammalian vertebrates, but low probabilities for non-primate mammals, consistent with a previous observation about the association of non-mammalian vertebrates with likely alignment artifacts[34].

**Conservation states positional enrichments**. Conservation states showed strong and distinct positional enrichments relative to annotated gene features including transcription start sites (TSS),

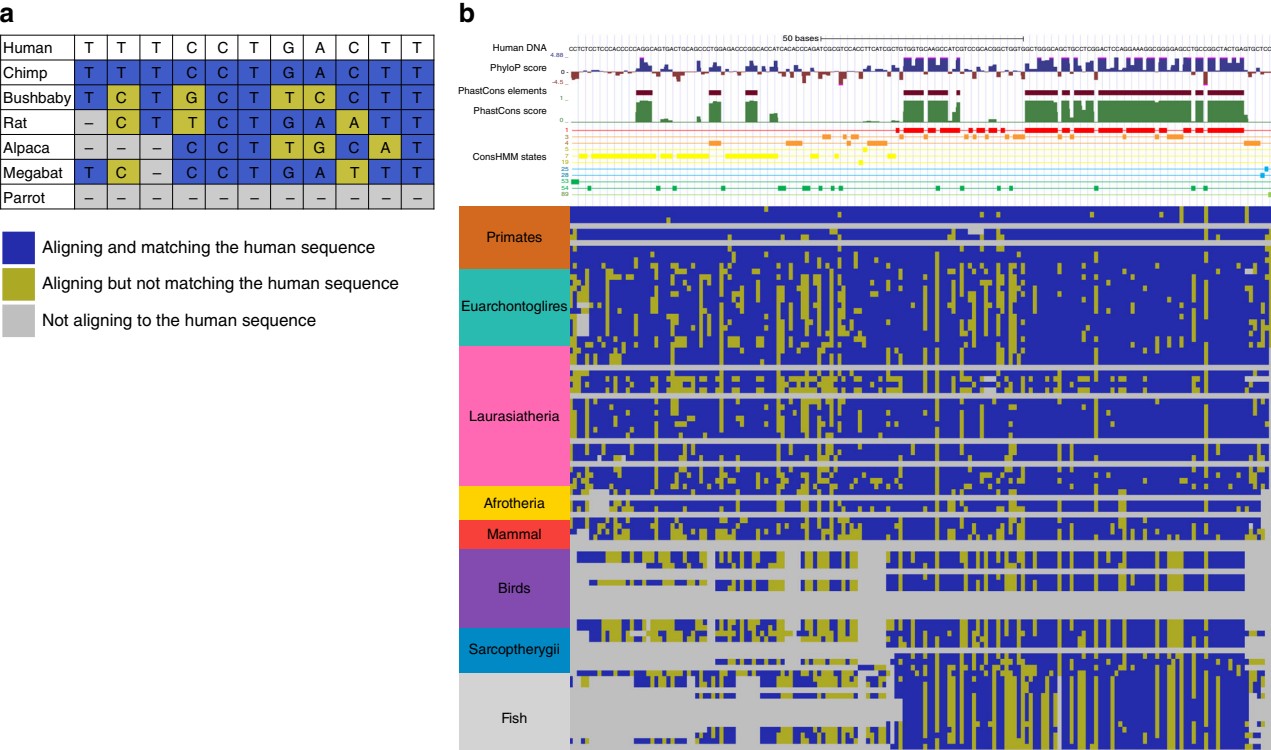

**Fig. 1** Illustration of ConsHMM modeling approach. **a** The input to ConsHMM is a multiple species alignment, which is illustrated for a toy example of 6 species aligned to the human sequence. At each position and for each species ConsHMM represents the information as one of three observations: (1) aligns with a non-indel nucleotide matching the human sequence shown in blue, (2) aligns with a non-indel nucleotide not matching the human sequence shown in yellow, or (3) does not align with a non-indel nucleotide shown in gray. **b** Illustration of conservation state assignments at the locus chr22:25,024,640-25,024,812 in hg19. Only states assigned to at least one nucleotide in the locus are shown. Below the conservation state assignments is a color encoding of the input multiple species alignment according to panel (**a**). The major clade of species as annotated on the UCSC genome browser[21] are labeled and ordered based on divergence from human. Above the conservation state assignments are PhastCons constrained elements and scores and PhlyoP constraint scores. This figure and Supplementary Fig. 10 together illustrate that positions of nucleotides that have the same status in terms of being in a constrained element or not or have similar constraint scores can be assigned to different conservation states depending on the patterns in the underlying multiple species alignment

transcription end sites (TES), and exon start and end sites, for both protein coding genes and pseudogenes. Within 20 base pairs (bp) of exon starts of protein coding genes, seven states (states 1–4, 7, 28, and 54) had at least 13-fold enrichment for some position, which also held for exons in specific coding phases (Fig. 3a, Supplementary Fig. 11a–c). These states were the only states that had a majority of positions aligning for at least some non-mammalian vertebrates, while still having a majority of positions aligning for all primates (Fig. 2a, Supplementary Data 1). Within exons, state 1 showed the strongest enrichment, consistent with its high matching probabilities through all vertebrates (Figs. 2b and 3a, b, Supplementary Fig. 11a–e). State 1 also had >40-fold enrichment at each of the three nucleotides immediately upstream of exon starts and six nucleotides downstream of exon ends (Fig. 3b, Supplementary Fig. 11c), corresponding to positions of the canonical 3′ and 5′ splice site sequences respectively, and consistent with their high conservation throughout vertebrates[35]. Downstream of the start of protein-coding exons, the enrichment profile for state 1 showed a 3-bp oscillation period, with a dip of enrichment at codon wobble positions. States 3 and 54 showed an inverse oscillation pattern, consistent with the states' high align probabilities through many vertebrates and lower match probabilities (Fig. 3a, Supplementary Fig. 11a–c).

Around the TSS of protein coding genes, state 28, which had moderate align and match probabilities for most vertebrates, had

the maximum enrichment (>30-fold) (Fig. 3c). Consistent with this enrichment, state 28 also had a 32-fold enrichment for CpG islands. However, state 28 was also 20-fold enriched for CpG islands >2 kb away from any TSS of protein coding genes and 10-fold enriched for TSS of protein coding genes >2 kb away from a CpG island. This suggests that both of these features are contributing to the association or the presence of unannotated TSS overlapping CpG islands[36]. Relative to TES of protein coding genes, enrichment of state 2, which had high align and match probabilities for almost all vertebrates except for fish, peaked at almost 12-fold (Supplementary Fig. 11f).

Relative to pseudogene exon starts and ends, states 100 and 82, both associated with alignability to distal vertebrates without many mammals closer to human (Fig. 2b, Supplementary Data 1), had enrichments peaking at greater than 100 and 38-fold respectively (Supplementary Fig. 11g, h). States 100 and 82 also showed the greatest enrichment relative to TSS of pseudogenes peaking at 184 and 68-fold respectively (Fig. 3d) and for TES of pseudogenes peaking at 199 and 61-fold respectively (Supplementary Fig. 11i).

Conservation states also had different positional enrichments relative to instances of regulatory motifs, with the enrichment varying at single nucleotide resolution (Fig. 3e, f, Methods)[37]. For example, states 2 and 5 reached 1.8-fold enrichments at some nucleotides in the *POU5F1* and *STAT* motifs respectively, but had lower enrichments (1.4–1.5) at other nucleotides with lower

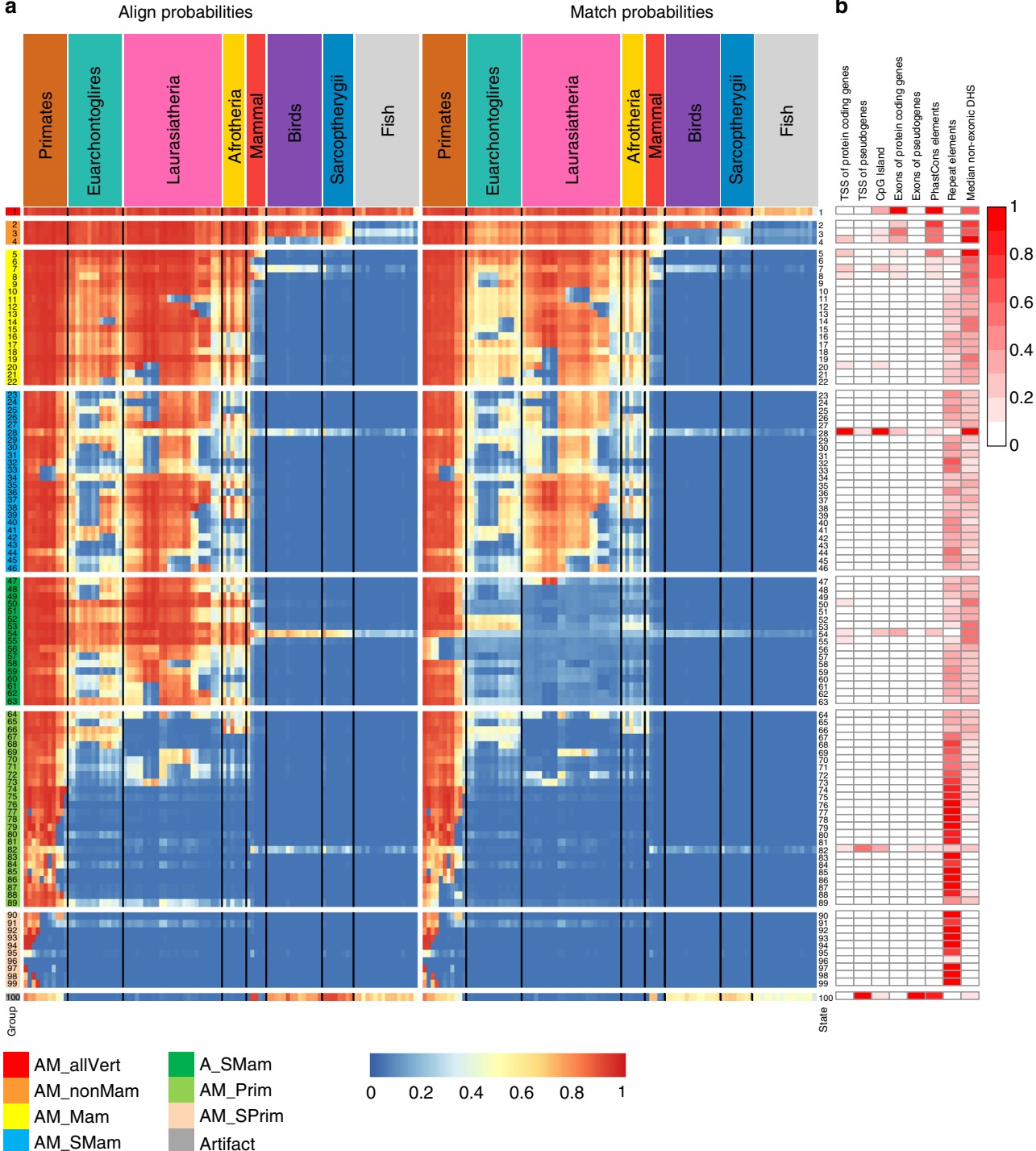

**Fig. 2** Conservation state emission parameters learned by ConsHMM and enrichments for other genomic annotations. **a** Each row in the heatmap corresponds to a conservation state. For each state and species, the left half of the heatmap gives the probability of aligning to the human sequence, which is one minus the probability of the not aligning emission. Analogously, the right half of the heatmap gives the probability of the matching emission. Each individual column corresponds to one species with the individual names displayed in Supplementary Fig. 5. For both halves, species are grouped by the major clades and ordered based on the hg19.100way.nh phylogenetic tree from the UCSC genome browser, with species that diverged more recently shown closer to the left[21]. The conservation states are ordered based on the results of applying hierarchical clustering and optimal leaf ordering[54]. The states are divided into eight major groups based on cutting the dendrogram of the clustering. The full dendrogram and an explanation of the group mnemonics is available in Supplementary Fig. 4. The groups are indicated by color bars on the left hand side and a white row between them. Transition parameters between states of the model can be found in Supplementary Fig. 6. **b** The columns of the heatmap indicate the relative enrichments of conservation states for external genomic annotations (Methods). For each column, the enrichments were normalized to a [0,1] range by subtracting the minimum value of the column and dividing by the range and colored based on the indicated scale on the right. Values for these enrichments and additional enrichments can be found in Supplementary Fig. 8 and enrichments for individual repeat classes and families can be found in Supplementary Fig. 14

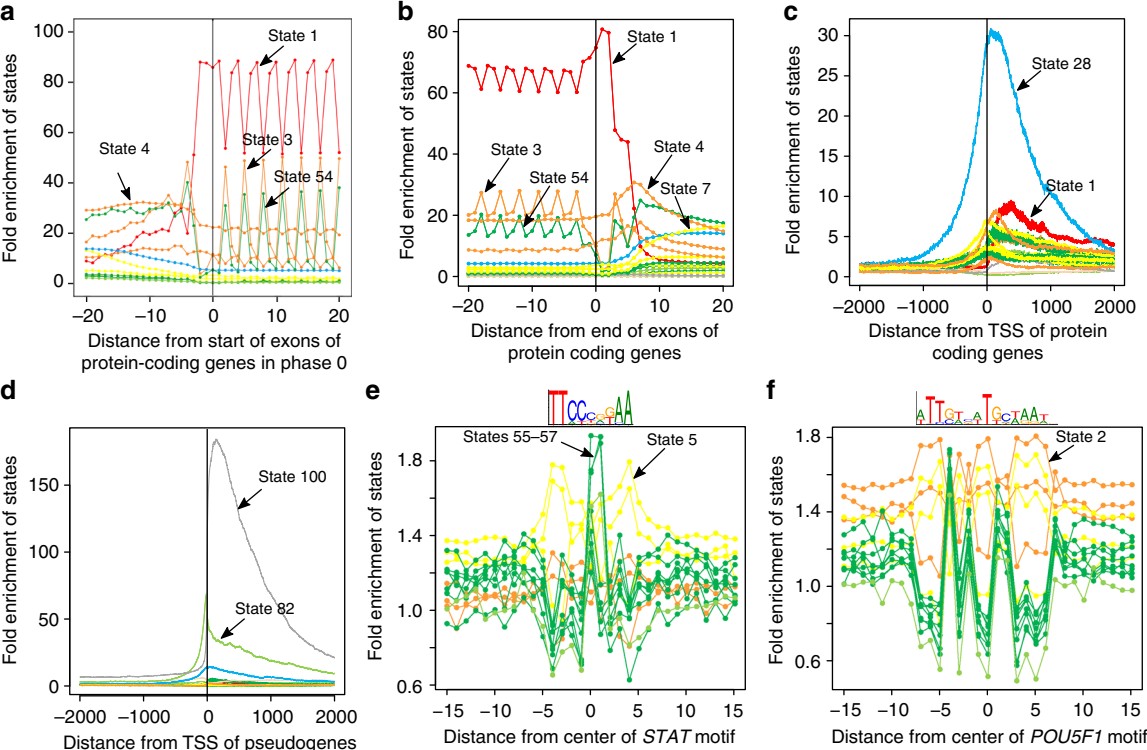

**Fig. 3** Conservation state positional enrichments. Plots of positional fold enrichments of conservation states relative to **a** start of exons of protein coding genes in phase 0, **b** end of exons of protein coding genes and **c**, **d** TSS of **c** protein coding, and **d** pseudogenes genes. Positive values represent the number of bases downstream in the 5′ to 3′ direction of transcription, while negative values represent the number of bases upstream. Enrichments relative to gene annotations are based on a genome-wide background. The subset of states included in panels (**a**)–(**d**) were the states that had at least a 3 fold enrichment at some position within ±2 kb from the anchor point. **e**, **f** Also shown are positional plots relative to the central nucleotide of a set of instances of **e** STAT and **f** POU5F1 motifs. The subset of states included in (**e**), (**f**) are the states that had an enrichment of at least 1.5 for some position within ±15 bp from the center nucleotide of either motif. Enrichments for motif instances were computed relative to the portion of the genome scanned for regulatory motifs in ref. [37], which excludes coding, 3′UTRs, and repeat elements. Additional position enrichment plots can be found in Supplementary Fig. 11

information content. States 55–57, which had high align probabilities for most mammals and low match probabilities even for most primates, peaked in enrichment at the CG dinucleotide in the center of the STAT motif, consistent with their genome-wide CG dinucleotides enrichments (Fig. 3e, Supplementary Fig. 12).

**Conservation state enrichments for different gene classes**. We next investigated conservation states enrichments for different gene classes. For each state, we determined the top 5% of gene promoter regions overlapping the state, which controls for different state preferences in general for promoters. For those corresponding genes, we evaluated Gene Ontology (GO) enrichments, which revealed distinct enrichment patterns (Fig. 4b, Supplementary Fig. 13, Methods). For example, states 1–3, which all had high alignability through at least birds, had substantial differences in their gene preferences. Out of these states, state 1 and state 3, which had high matching through all vertebrates and mainly mammals respectively, were the only ones enriched for nucleosomes ($p < 10^{-41}$; 10.5-fold) and sensory perception of smell genes ($p < 10^{-300}$; 15.5-fold) respectively. State 2, which had high match probabilities through all vertebrates except fish, was the state most enriched for cellular developmental processes ($p < 10^{-30}$; 1.8-fold), which were not enriched in state 3. States with overall lower align or match probabilities also had notable enrichments. For example, state 89, which had moderate alignability for most non-primate mammals, but low matching even for primates, was the state most enriched

for antigen binding ($p < 10^{-14}$; 6.7-fold) consistent with antigen binding being associated with many species, but fast evolving[38].

**Conservation state enrichments for repeat elements**. The conservation state enrichments for bases in repeat elements ranged widely from twofold enrichment to 133-fold depletion (Fig. 2b, Supplementary Fig. 8)[21,39]. Of the 25 states in which only primate species had a majority of positions aligning, all but states 89 and 96 had an enrichment of 1.55 or greater for repeat elements, while the other 75 states all had a lower enrichment or were depleted (Supplementary Data 1). Neither state 89 nor 96 enriched for repeat elements. As noted above, state 89 is associated with fast evolving bases shared with some non-primate mammals, while state 96 is associated with assembly gaps (Supplementary Fig. 8).

Individual conservation states had distinct enrichments for different repeat classes (Supplementary Fig. 14). For instance, different states had maximal enrichments for the DNA, LINE, LTR, and SINE repeat classes and families (Fig. 4d). State 74, which had high align and match probabilities for all primates, had the maximal enrichment of 5.6-fold for DNA repeats, while the enrichment for the other three classes were between 1.0 and 1.8-fold. State 86, which lacked alignability of a subset of primates, had the maximal of 3.0-fold enrichment for LINE repeats, while the enrichment for the other classes were between 0.6 and 1.6-fold. States 76 and 77 had maximal enrichments of 3.3 and 4.5-fold for LTR and SINE respectively compared to 1.1 and 2.1-fold for SINE and LTR respectively. States 76 and 77 both had high align probabilities through primates up to and including squirrel

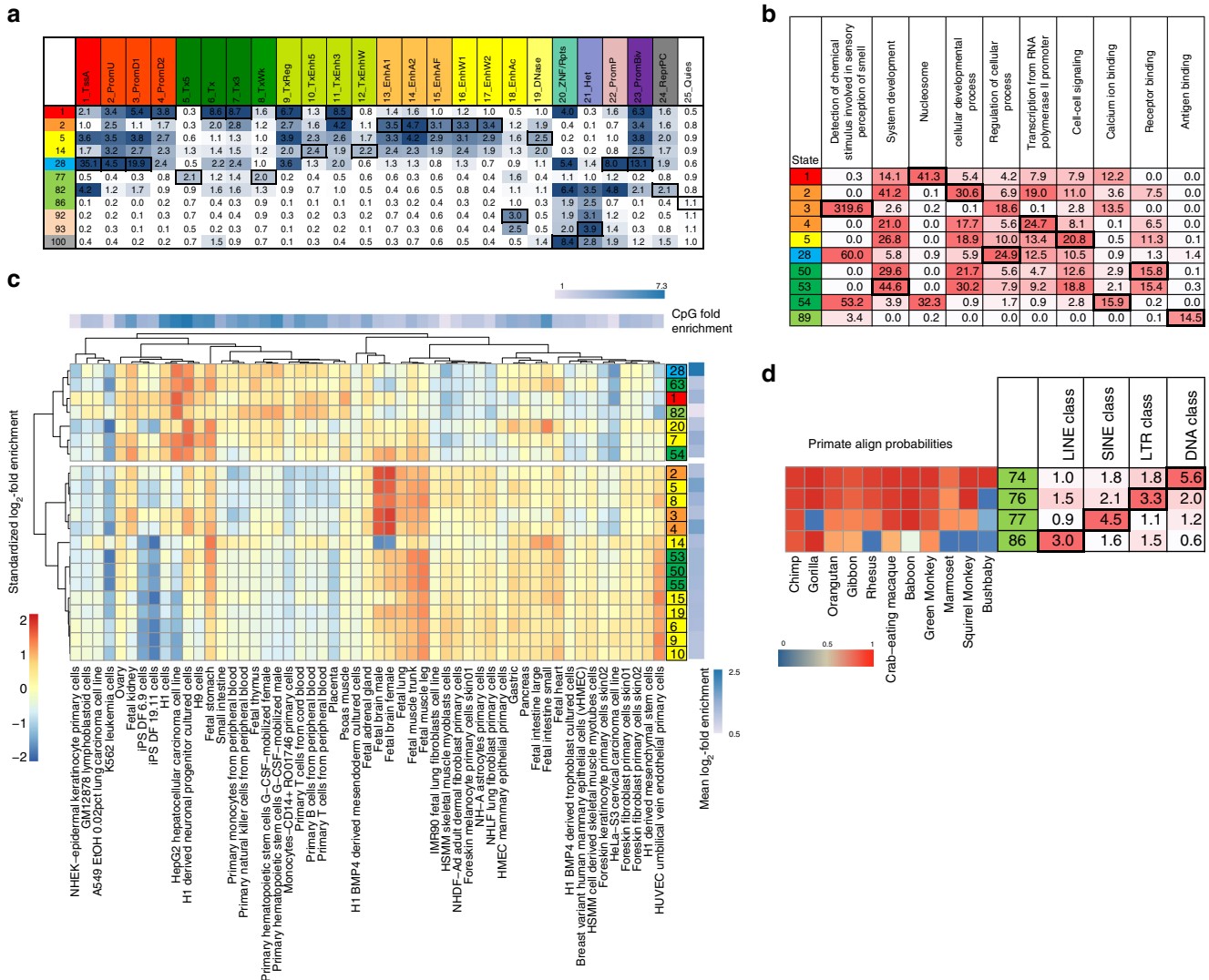

**Fig. 4** Conservation states enrichment for chromatin states, GO terms, DHS and repeat elements. **a** Median fold enrichment of conservation states (rows) for one of 25 chromatin states from a previously defined chromatin state model defined across 127 samples of diverse cell and tissue types (columns)[40]. Only conservation states that had the maximum value for at least one chromatin state are shown, and those values are boxed. See Supplementary Fig. 15 for the enrichments of all conservation states. **b** −log10 p-value (uncorrected) of the conservation states (rows) for the GO term (columns) where each conservation state is associated with its top 5% genes based on promoter regions (Methods). Only GO terms which were the most significantly enriched term for some conservation state among terms the state was maximally significant for are shown, restricted to the top 10 terms based on the significance of the enrichment. Only conservation states that had the most significant enrichment for one of the displayed GO terms are shown, with the maximal enrichments boxed. The full set of conservation states with additional GO terms are in Supplementary Fig. 13. **c** Relative enrichments of conservation states for DHS across cell and tissue types. Only conservation states with at least a twofold enrichment in one sample considered are shown. Enrichment values were log2 transformed and then row normalized by subtracting the mean (right heatmap) and dividing by the standard deviation. States and experiments were then hierarchically clustered and revealed two major state clusters. In the top cluster conservation states showed the greatest enrichment for experiments in which the DHS also strongly enriched for CpG islands (top heatmap). In the bottom cluster conservation states had the strongest relative preference for fetal related samples or HUVEC. **d** Fold enrichment of conservation states with the maximal enrichment for LINE, SINE, LTR or DNA repeats next to the state align probabilities for primates. These states all had low align probabilities outside of primates, but their differences among primates corresponded to substantial differences in repeat enrichments[28].

monkey, with the exception that state 77 lacked alignability to gorilla. Despite these subtle differences in alignment probabilities, these states had substantial differences in their repeat enrichments.

**Relationship of conservation states to chromatin states**. To understand the relationship of conservation states to chromatin states we determined the median enrichment of each conservation state for 25-chromatin states defined across 127 samples using imputed data[5,40] (Fig. 4a, Supplementary Fig. 15). Eleven

conservation states were maximally enriched for at least one of the chromatin states. Conservation state 28 had the greatest enrichment for any chromatin state, with a 35-fold enrichment for an active promoter chromatin state, and was maximally enriched for four other promoter associated chromatin states. Conservation state 1 was maximally enriched (3.8–8.7-fold) for five chromatin states associated with transcribed and exonic regions[40], consistent with its maximal enrichment for annotated exons. Conservation state 2 was maximally enriched (3.1–4.7-fold) for five enhancer associated chromatin states, while

conservation state 5 had high enrichments for these states and was maximally enriched (2.5-fold) for a chromatin state primarily associated with just signal of DNase I hypersensitive sites (DHS). These chromatin state enrichments highlight the multidimensional information that conservation states capture.

**Conservation states and cell type specific DHS**. We next investigated whether different conservation states capture distinct enrichment patterns for DHS across cell and tissue types. We analyzed DHS from the 53 samples considered above for which maps of experimentally observed DHS were available[5]. We hierarchically clustered the row normalized enrichment patterns of the 21 conservation states that exhibited at least twofold enrichment in one or more samples, revealing two major clusters of states (Fig. 4c). One major cluster contained 14 states, with ten of the states having maximum enrichment for a fetal sample and the remaining four states having maximum enrichment for the cell type Human Umbilical Vein Endothelial Cells (HUVEC). The second major cluster consisted of seven states, all of which were enriched for CpG islands (Fig. 2b, Supplementary Fig. 8). The samples for which DHS had the greatest enrichments for states in this cluster also had the greatest enrichment for CpG islands (Fig. 4c, Methods), but were biologically diverse in the type of cell or tissue and could potentially reflect technical differences.

**Conservation states' relationship to constraint annotations**. We next investigated the relationship of the conservation state annotations with constrained element sets from four methods (GERP++, SiPhy-omega, SiPhy-pi, and PhastCons) and univariate scores of evolutionary constraint from three methods (GERP++, PhastCons, and PhyloP). The PhastCons and PhyloP constraint annotations were defined on the same alignment as the conservation states. The available GERP++, SiPhy-omega, and SiPhy-pi constraint annotations were defined from different versions of Multiz alignments and only considered mammals.

States 1–5 all had >9.0-fold enrichment for each constrained element set and high mean constraint scores consistent with their high matching probabilities across all mammals (Fig. 2b, Supplementary Fig. 16). States 54 and 100 also had >6.0-fold enrichment for at least one constrained element set. State 100, which had high aligning and matching primarily in non-mammalian vertebrates, had 15-fold enrichment for PhastCons elements and high mean PhastCons and PhyloP scores, consistent with these scores being defined using non-mammalian vertebrates. State 54, which had high alignability through most vertebrates and low matching outside primates, enriched 4 to 7-fold for the constrained element sets, but did not show high mean base-wise scores particularly for the GERP++ and PhyloP scores, consistent with its enrichments for codon wobble positions. More generally, constrained element sets, except for PhastCons, did not show biologically relevant variation at single nucleotide resolution in their enrichments around regulatory motifs and exon start and ends as the conservation state annotations did (Fig. 3a, e, f, Supplementary Fig. 17).

We compared biologically relevant information in conservation state and constraint annotations using established genome annotations. We evaluated their ability to recover annotated TSS, TES, and exon starts and ends separately for protein coding and pseudogenes (Fig. 5a–c, Supplementary Fig. 18). In almost all cases the conservation states provided greater information for recovering annotated gene features. The only exceptions were that PhyloP scores had higher precision at low recall levels for protein coding exon starts and ends, and that SiPhy-pi elements had slightly higher precision for TSS of protein coding genes at their one recall point.

We also evaluated recovering bases covered by DHS (Supplementary Figs. 19 and 20, Methods). When comparing DHS recovery from 53 samples in aggregate, the conservation states had greater precision at the same recall level than all the constraint scores and PhastCons elements, both genome-wide and for non-exonic bases. The precision for GERP++, SiPhy-pi and SiPhy-omega elements was higher at their single recall point (Supplementary Fig. 19). Similar results were seen for regions distal to TSS, except for some scores at low recall levels in the non-exonic comparison. The higher precision for GERP++, SiPhy-pi and SiPhy-omega elements in the aggregate evaluation over constraint scores, PhastCons elements, and conservation states might be related to the coarser resolution at which they were defined and also did not hold for all cell types (Supplementary Figs. 17 and 20).

Conservation states also had complementary information about DHS to constrained elements, as constrained element enrichments for DHS varied substantially depending on their conservation state (Fig. 5d, Supplementary Figs. 21 and 22). For example, PhastCons elements' bases in 35 states were depleted for Fetal Brain DHS in non-exonic regions, covering 10% of PhastCons bases, while PhastCons elements' bases in 12 states bases were enriched over fivefold, covering 37% of PhastCons bases. Additionally, bases not in a constrained element in some states had greater enrichments for DHS than bases in a constrained element in other states. Constrained elements also offered additional information, as in most cases bases that were in a constrained element in a given conservation state had greater enrichment for DHS than those that were not.

We also analyzed conservation state enrichments for previously defined subsets of PhastCons constrained non-exonic elements (CNEEs) based on a directed phylogenetic approach that assigned each element to a phylogenetic branch point of origin[22] (Supplementary Fig. 23a). Bases in elements assigned to the Tetrapod clade branch point of origin had a 37-fold enrichment for state 2, which had high aligning and matching through all vertebrates except fish, but also 51-fold enrichment for state 100, associated with likely alignment artifacts, demonstrating the heterogeneous nature of assignments from directed phylogenetic partitioning. We also evaluated the subsets of CNEEs enrichment for CpG islands within non-exonic regions (Supplementary Fig. 23c). The most enriched subset of CNEEs was 6.7-fold enriched covering 1.9% of non-exonic CpG islands. In comparison, conservation state 28 had a 37.6-fold enrichment, while covering 12.8% of such bases. A similar pattern of enrichments was observed when only considering CNEEs overlapping a PhastCons element called on the same alignment as the conservation states (Supplementary Fig. 23b, d). These results highlight that the conservation states capture additional biological information compared to directed phylogenetic based approaches.

**Conservation states enrichments for prioritized variants**. Various scores have been proposed to prioritize variants, including based on inter- or intra-species constraint or integration of diverse genomic annotations. However, a systematic understanding of different types of bases these scores prioritize is generally lacking. To address this, we analyzed conservation states' genome-wide enrichments of top 1, 5, and 10% prioritized bases by 12-different scores (CADD (v1.4), CDTS, DANN, Eigen, Eigen-PC, FATHMM-XF, FIRE, fitCons, GERP++, PhastCons, PhyloP, and REMM). We also analyzed the enrichment specifically in non-coding regions for those scores and two non-coding only scores, LINSIGHT and FunSeq2 (Fig. 6a, b, Supplementary Figs. 24–27)[8,9,13,16,18,19,41–47].

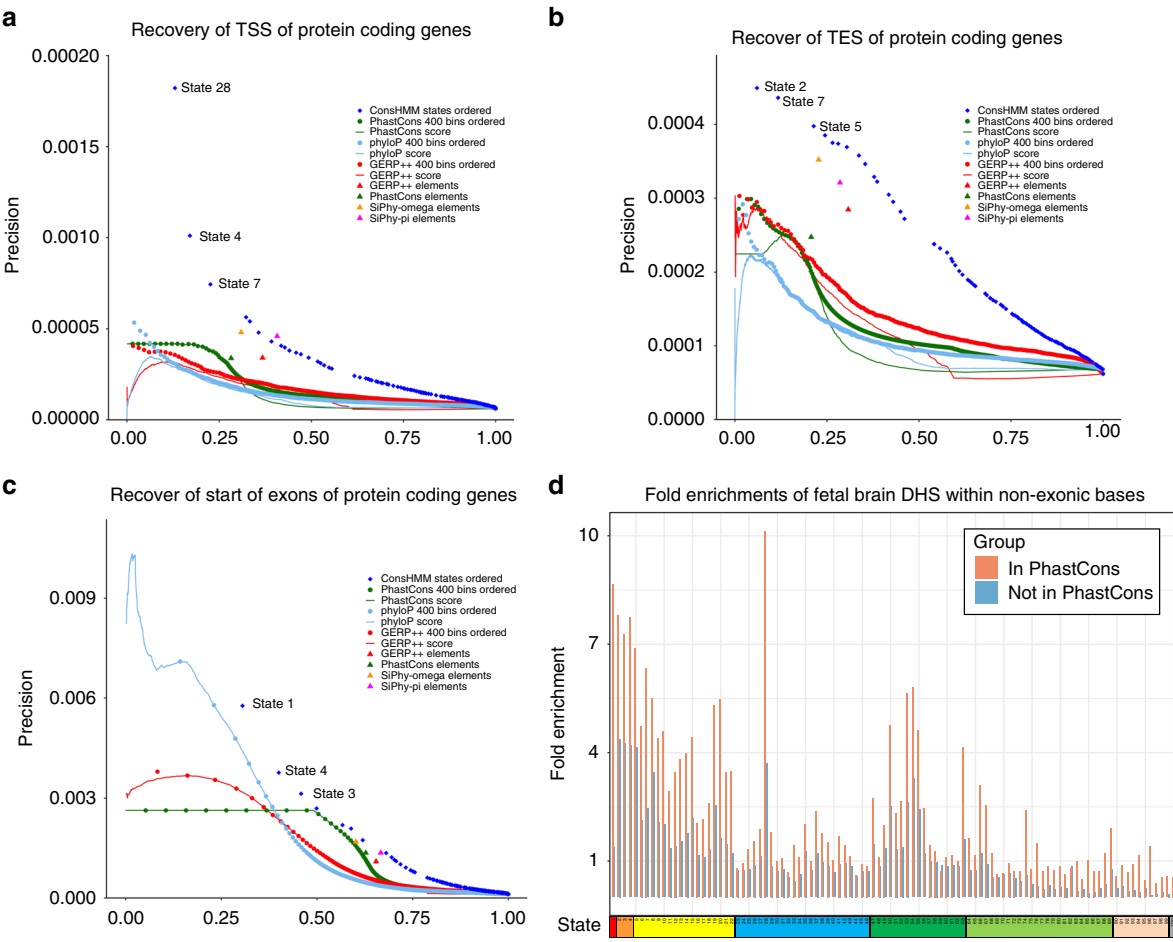

**Fig. 5** Relationship of conservation states with constrained elements and scores. Precision-recall plots for recovery of **a** TSS of protein coding genes, **b** TES of protein coding genes, and **c** the start of exons of protein coding genes. Recovery based on ordering ConsHMM conservation states for their enrichment for the target set in the training data, then cumulatively adding the states in that ranked order and evaluating on the test data is shown with a series of blue dots (Methods). The first few conservation states added are labeled with their state number. Recovery based on ranking from highest to lowest value of constraint scores is shown with continuous lines. Recovery based on score partitioning into 400 bins and subsequent ordering based on enrichment for the target set in the training data, then cumulatively adding bins in that ranked order and evaluating on the test data is shown in a series of dots of the same color as the continuous line corresponding to the score. Recovery of target test bases by a constrained element set is shown with a single dot for each constrained element set. See Supplementary Figs. 18–20 for plots based on additional targets. **d** The graph shows the fold enrichment for Fetal Brain DHS[5] within the non-exonic portion of each conservation state, separately for those bases in a PhastCons constrained element (pink) and bases not in such an element (blue). Enrichments within constrained elements varied substantially depending on the conservation state. For a given conservation state, bases in a constrained element had greater enrichments than bases not in a constrained element, illustrating complementary information of conservation states and constrained elements. See Supplementary Fig. 21 for graphs based on different element sets or DHS data and Supplementary Fig. 22 for these enrichments plotted against the size of the set

Bases prioritized by most scores had strong enrichments for specific conservation states. For example, state 1, which had high align and match probabilities across all vertebrates, had a 77.2-fold enrichment for CADD top 1% prioritized bases genome-wide, covering 46% of such bases. Despite the CADD score being based in part on many non-conservation annotations, this enrichment was greater than that observed for any inter-species constraint score. There was a general consistency in states with higher enrichment across the various measures. For example, in top 1% bases for the genome-wide analysis, only 13 states were among the top five most enriched by at least one of the 12 scores. Nine of these 13 states (states 1–5, 7, 28, 54, 100) were in the top five for at least three scores. However, there were also important enrichment differences between scores for these states, and in several cases a single score prioritized other states.

There was substantial disagreement among the scores of the relative importance of states 2 and 28, the most enhancer and

promoter enriched states respectively, particularly in non-coding regions. For example, state 2 was the second or third most enriched state (24.9–47.2-fold) for CADD, Eigen, FATHMM-XF, GERP++, LINSIGHT, PhastCons, PhyloP, and REMM top 1% prioritized bases in non-coding regions. On the other hand, state 28 had lower enrichments (0.3 to 6.2-fold) and was not one of the top five most enriched states for any of those scores. In contrast, for CDTS, DANN, and Eigen-PC, state 28 was the first or second most enriched state (7.6–18.6-fold), while state 2 had lower enrichments (0.8–2.1-fold) and was not among the top five most enriched states.

There was a large disagreement in the state enrichments between variants prioritized by DANN and CADD for both the current and original versions of CADD (Supplementary Figs. 25–27). This was despite DANN using the same framework as CADD except using a deep neural network[43]. Surprisingly, for top 1% non-coding variants, DANN showed a depletion for state 2, which had high

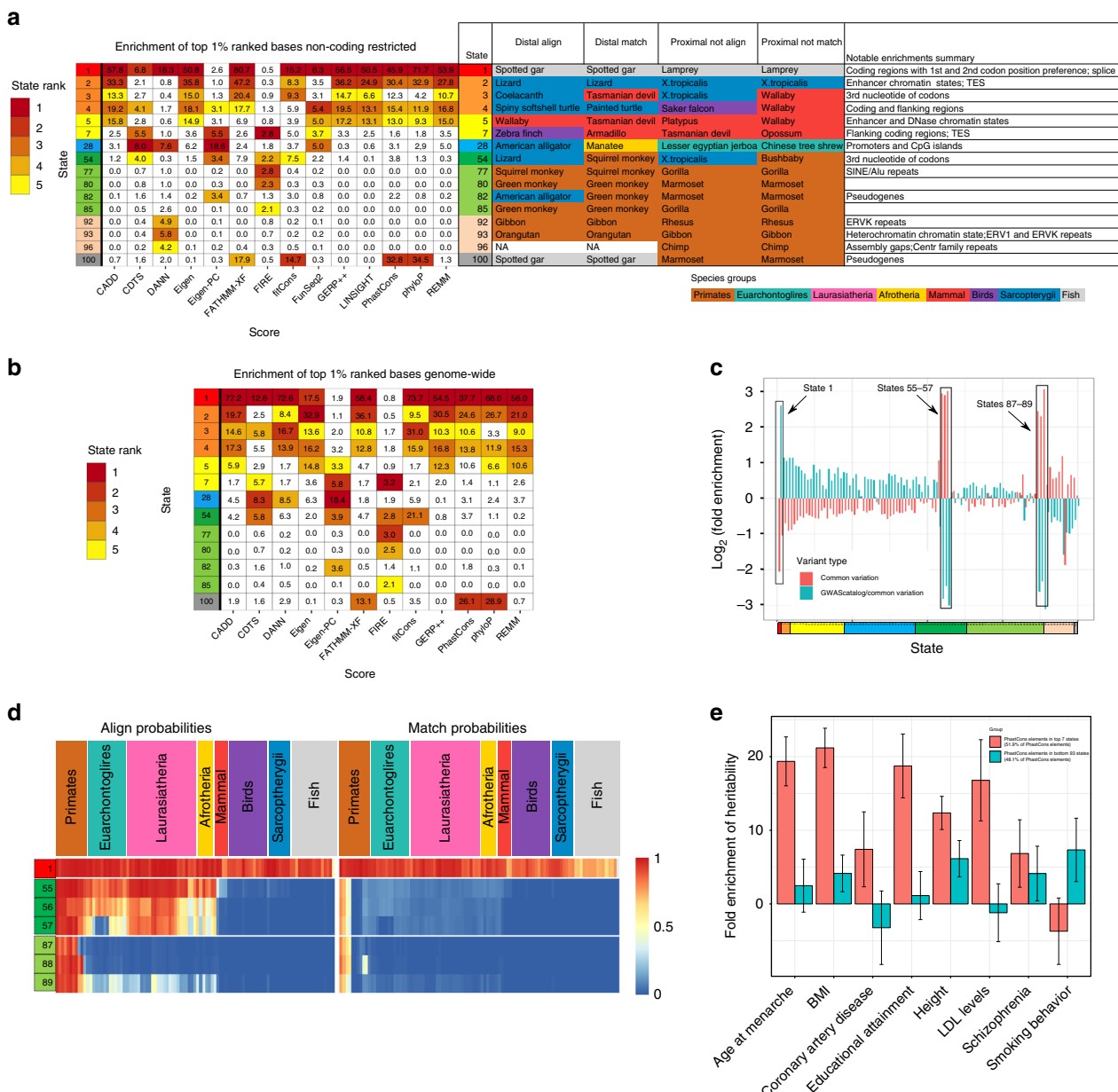

**Fig. 6** Conservation states' association with human genetic variation. **a** Fold enrichments of bases ranked in the top 1% of the non-coding genome by 14 variant prioritization scores. Only states among the top five most enriched states for at least one score are shown. The enrichment of the top five ranking states for each score is colored according to their ranking. The table provides a summary of the align and match probabilities and notable enrichments of each state. The 'Distal align' and 'Distal match' columns contain the species most distal to human that has an alignment and matching probability in the state >0.5, respectively. The 'Proximal not align' and 'Proximal not match' columns contain the species closest to human that has an alignment and matching probability in the state lower than 0.5, respectively. The species are colored by the major clades indicated below. An expanded version including all states is available in Supplementary Data 1. **b** Enrichments of bases ranked in the top 1% genome-wide by 12 variant prioritization scores. The criteria for selecting states to display and coloring enrichments was the same as panel (**a**). Enrichments for prioritized bases at additional thresholds and for all states both genome-wide and for the non-coding genome are in Supplementary Figs. 24–27. **c** The $\log_2$ fold enrichment of each state for common SNPs (pink) and GWAS catalog variants relative to common SNPs (blue). **d** The representation of state emission parameters from Fig. 2a for the subset of states highlighted in panel (**c**). **e** Heritability partitioning enrichments from the method of ref. [15] applied on two disjoint subsets of bases in PhastCons elements, with eight phenotypes previously analyzed with heritability partitioning in the context of a baseline annotation set (Methods). The two sets are PhastCons elements overlapping one of the seven conservation states showing the greatest enrichment for DHS in its non-exonic portion (states 1–5, 8, and 28) covering 51.9% of PhastCons bases (pink) and bases in PhastCons elements overlapping the remaining 93 states covering 48.1% of PhastCons bases (blue). Error bars represent standard errors around the enrichment estimate using jackknife resampling

matching probabilities through all vertebrates except fish, while having over four-fold enrichment for multiple states that showed high alignment or matching probabilities for only subsets of primates.

There were also notable enrichment differences for other states for which the biological importance was less apparent. For example, state 100, associated with likely alignment artifacts, in the top 1% non-coding region analysis had enrichments in the

range of 14.7 to 34.5-fold for FATHMM-XF, fitCons, PhastCons and PhyloP prioritized bases, while the enrichment for all other scores was at most 2.0-fold. Another example was state 54, which associated with wobble position within codons, and had a 21.1 fold enrichment in the top 1% genome-wide analysis for fitCons prioritized bases and was also the third most enriched state for CDTS, Eigen-PC, and FIRE, while depleting for GERP++ and REMM prioritized bases. These results highlight how the conservation states enable recognizing and characterizing distinct subsets of nucleotides that are selectively captured by different variant prioritization scores.

**Conservation states and human genetic variation**. Previous analyses have found a depletion of human genetic variation in evolutionarily constrained elements[7]. Consistent with that, the greatest depletion (3.3-fold) of common single nucleotide polymorphisms (SNPs) is in state 1, the state most enriched for constrained elements, while states 55–57 and 87–89 had the greatest enrichments for common SNPs (5–8-fold). These six states all had high align, but low match probabilities for most primates and had the greatest enrichment of CG dinucleotides (Supplementary Fig. 12). We observed similar patterns of enrichments and depletions for variants identified from whole genome sequencing[47], with their magnitude increasing with minor allele frequency (Supplementary Fig. 28).

States had opposite enrichment patterns for GWAS catalog variants[48] relative to the background of common SNPs (Fig. 6c, d). Using this background, state 1 was most enriched for GWAS catalog variants, consistent with constrained elements enriching for GWAS variants[7]. States 55–57 and 87–89 showed the greatest depletion, suggesting that a variant in one of these states is less likely to be phenotypically associated.

We also applied the INSIGHT[49] model to obtain its estimates of the density of positive selection events and percentage of bases under selection within human populations in each conservation state (Supplementary Fig. 29). States 54–57 and 87–89 all had substantial density for positive selection event estimates. INSIGHT also estimated that 77% of states had more than 75% bases under-selection, while 13% had <50% bases under selection. Similar estimates held when restricting to bases in PhastCons elements and not in PhastCons elements (Supplementary Fig. 29). However, instead of a majority of states actually having a high percentage of bases under selection, this likely reflects that there is a relatively direct relationship between human variation information contained by the conservation states and INSIGHT's use of such information to quantify selection.

**Conservation states and heritability partitioning**. Previous analyses have suggested strong enrichments of constrained elements and DHS for phenotype heritability[15,50]. Given the differences in DHS enrichments of constrained elements across conservation states, we investigated whether constrained elements in conservation states most enriched for DHS had different phenotype heritability than those in other states. Specifically, we ranked the conservation states in descending order of their median enrichment within non-exonic bases for DHS from 123 experiments (Fig. 2b, Methods)[3]. We then partitioned bases in PhastCons elements into two almost equal size sets based on whether they overlapped a top seven-ranked conservation state (states 1–5, 8, 28). We computed the heritability for the two sets for eight phenotypes in the context of baseline annotations that include DHS[15]. For seven of the phenotypes, bases in constrained elements overlapping the top seven states had greater enrichment than those in the other states, often substantially so (Fig. 6e).

These results suggest possible additional value of conservation states for isolating disease-associated variants.

**Discussion**

We introduced the ConsHMM method for genome annotation and used it to annotate the human genome at single nucleotide resolution into one of 100 conservation states. ConsHMM learns conservation states de novo using a multivariate HMM based on the combinatorial and spatial patterns of which species align and match a reference genome in a multi-species DNA sequence alignment. Conservation states had substantial enrichments for a wide range of other genomic annotations, functional genomics data, and human variation data.

ConsHMM differs from other commonly used comparative genomics based annotation approaches in several respects. One difference is that it takes an unsupervised approach that does not explicitly use a phylogenetic tree in its modeling. This leads to relatively unbiased, flexible and interpretable models. Despite not explicitly using a phylogenetic tree, many state patterns discovered are consistent with commonly assumed phylogenetic relationships of the species. While states' parameters often decreased with divergence time from human, there were some exceptions. Some of these exceptions corresponded to missing specific sub-clades of species, particularly those with long branch lengths. For example, in some states mouse and rat were absent, while more distally diverged mammals were present. Other states isolated likely artifacts in alignments that heavily enriched for pseudogenes. A second difference is that ConsHMM explicitly differentiates non-aligning bases from aligning non-matching bases, which allowed it, for example, to identify states such as those associated with third codon positions. A third difference between the ConsHMM annotations and standard constraint measures is that the ConsHMM annotations are defined directly relative to the variant present in the genome being annotated. When applying ConsHMM to annotate the human genome, a mutation unique to human would be expected to have a much larger effect on the ConsHMM annotations than a mutation unique to a single other species. This would not in general be expected for constraint measures that treat the target genome for annotation in the same way as other genomes in an alignment. An interesting future direction would be to produce and analyze individual specific ConsHMM annotations.

ConsHMM annotations are complementary to existing binary elements and scores of evolutionary constraint based on phylogenetic modeling. Both bases within and outside of constrained elements are heterogeneous in their assigned conservation states. ConsHMM annotations provide additional information about the conservation patterns at each base. In many cases, the conservation states had greater information than constraint scores or elements for predicting external annotations. Notably, ConsHMM identified a conservation state strongly enriched for TSS and CpG islands that was not well captured by phylogenetic modeling approaches. For other annotations, such as DHS, the relative information depended on the constrained element set or score being compared. Importantly, the DHS information provided by the states was complementary to information in the constrained elements. Furthermore, we observed that bases in constrained elements showed substantially different enrichments for phenotype-associated heritability, depending on their conservation state. The conservation state annotations also provide a useful framework for understanding the types of bases prioritized by constraint scores or other types of variant prioritization scores, since the corresponding conservation patterns are defined systematically in an unbiased way, at single nucleotide resolution and capture a diverse set of biological features.

ConsHMM is both inspired by, and provides complementary information to, ChromHMM. While the annotations produced by the two methods have fundamental differences, they also exhibited substantial cross-enrichments. In general, conservation states have the advantages of providing information at single nucleotide resolution and about bases active in cell types that have not been experimentally profiled, while chromatin states have the advantage of directly providing cell type specific information.

We expect many applications for the ConsHMM method and annotations. The ConsHMM method can be readily applied to alignments to other reference species or alignments by other methods[26]. The ConsHMM annotations are a resource to interpret other genomic datasets or variant prioritization scores. A possible avenue for future work would be to integrate the conservation states with other genomic annotations to produce a variant prioritization score. An effective strategy for that would need to be powered to retain the rich information in the conservation state annotations, and would also need to be based on a principle sufficiently independent from how the conservation states are defined to enable a meaningful integration and prioritization. This work represents a step towards improving whole genome annotations, including of non-coding regions and variants, which will be of continued importance towards understanding disease.

## Methods

**Modeling conservation states with ConsHMM.** ConsHMM takes as input an $N$-way multi-species sequence alignment to a designated reference genome. For each base in the reference genome, $i$, ConsHMM encodes information from the multiple species alignment into a vector, $v_i$, of length $N-1$. An element of the vector, $v_{i,j}$, corresponds to one of three possible observation for a non-reference species $j$ at position $i$. The three possible observations are: (1) the non-reference species aligns with a non-indel nucleotide symbol present matching the reference nucleotide, (2) the non-reference species aligns with a non-indel nucleotide symbol present, but does not match the reference nucleotide, or (3) the non-reference species does not align with a non-indel nucleotide symbol present.

ConsHMM assumes that these observations are generated from a multivariate HMM where the emission parameters are assumed to be generated by a product of independent multinomial random variables, corresponding to each non-reference species in the alignment. Formally, the model is defined based on a fixed number of states $K$, and number of species in the multiple sequence alignment $N$. For each state $k$ ($k = 1, \ldots, K$), non-reference species $j$ ($j = 1, \ldots, N-1$) and possible observation $m$ ($m = 1, 2,$ or $3$ as described above), there is an emission parameter: $p_{k,j,m}$ corresponding to the probability in state $k$ for species $j$ of having observation $m$. For each possible observation $m$, let $I_m(v_{i,j}) = 1$ if $v_{i,j} = m$, and 0 otherwise. Let $b_{t,u}$ be a parameter for the probability of transitioning from state $t$ to state $u$. Let $c \in C$ denote a chromosome, where $C$ is the set of all chromosomes in the reference genome of the multiple species alignment, and let $L_c$ be the number of bases on chromosome $c$. Let $a_k$ ($k = 1, \ldots, K$) be a parameter for the probability of the first base on a chromosome being in state $k$. Let $s_c \in S_c$ be a hidden state sequence on chromosome $c$ and $S_c$ be the set of all such possible state sequences. Let $c_h$ denote position $h$ on chromosome $c$. Let $s_{c_h}$ denote the hidden state at position $c_h$ for state sequence $s_c$.

We learn a setting of the model parameters that aims to optimize

$$P(v|a,b,p) = \prod_{c \in C} \sum_{s_c \in S_c} a_{s_{c_1}} \left( \prod_{i=2}^{L_c} b_{s_{c_{i-1}}, s_{c_i}} \right) \prod_{h=1}^{L_c} \prod_{j=1}^{N-1} \prod_{m=1}^{3} p_{s_{c_h,j,m}}^{I_m(v_{c_h,j})}$$

Once a model is learned, each nucleotide is assigned to the state with maximum posterior probability. To conduct the model learning and state assignments, ConsHMM calls an extended version of the ChromHMM[31] software, originally designed to solve an analogous problem of annotating a genome into chromatin states based on combinatorial and spatial patterns of the presence of different chromatin marks. The modeling in ConsHMM differs from the typical use of ChromHMM in three main respects: (1) the observation for each feature comes from a three-way multinomial distribution as opposed to a Bernoulli distribution, (2) it is applied at single nucleotide resolution as opposed to 200-bp resolution, (3) it is applied with more features than ChromHMM models have used in the past. (2) and (3) raise scalability issues in terms of time and memory, which we addressed in an updated version of ChromHMM (see below).

To apply ChromHMM in the context of three-way multinomial distributions, ConsHMM represents the three possible observations at position $i$ for a species $j$ with two binary variables, $y_{ij}$ and $z_{ij}$, corresponding to aligning and matching the reference genome respectively. $y_{ij}$ has the value of 1 if the other species aligns to the

reference with a non-indel nucleotide and 0 otherwise. $z_{ij}$ has the value of 1 if the other species has the same nucleotide as the reference sequence and has a value of 0 if the other species has a different nucleotide present than the reference. In the case in which $y_{ij} = 0$, there is no nucleotide to compare to the reference and that value of the $z_{ij}$ variable is considered missing (encoded with a '2' for ChromHMM). If the value of an observed variable is missing, ChromHMM excludes the Bernoulli random variable corresponding to the observation from the emission distribution calculation at that position. For each state $k$ and species $j$, ChromHMM thus learns two parameters, $f_{k,j}$ and $g_{k,j}$. $f_{k,j}$ corresponds to the probability that at a given position in state $k$, species $j$ aligns to the reference genome with a non-indel nucleotide, that is $P(y_{i,j} = 1| s_i = k)$. $g_{k,j}$ corresponds to the probability that at a given position in state $k$, species $j$ matches the reference genome conditioned on species $j$ aligning with a non-indel nucleotide, that is $P(z_{i,j} = 1 | y_{i,j} = 1$ and $s_i = k)$. This representation is equivalent to the three-way multinomial distribution, ($p_{k,j,1}$, $p_{k,j,2}$, $p_{k,j,3}$) described above where $p_{k,j,1} = P(y_{i,j} = 1, z_{i,j} = 1 \mid s_i = k)$, $p_{k,j,2} = P(y_{i,j} = 1, z_{i,j} = 0 \mid s_i = k)$, and $p_{k,j,3} = P(y_{ij} = 0 \mid s_i = k)$, since $p_{k,j,1} = f_{k,j} \times g_{k,j}$, $p_{k,j,2} = f_{k,j} \times (1-g_{k,j})$, and $p_{k,j,3} = 1 - f_{k,j}$.

**Multiple species sequence alignment choice.** ConsHMM can be applied to any multiple species sequence alignment which is available in multiple alignment format (MAF) or which can be converted into this format. For the results presented here we applied it to the 100-way Multiz vertebrate alignment with human (hg19) as the reference genome[21,33] for chromosomes 1–22, X, and Y.

**Scaling-up ConsHMM to single base resolution.** Since for our application ConsHMM needs to run ChromHMM at single base resolution ('-b 1' flag) with 198 features after our binary encoding (two for each non-human species in the 100-way alignment), we had to address scalability issues in terms of both memory and time. To address the memory issue we modified ChromHMM to support only loading in main memory input for chromosome files it is actively processing, as previously ChromHMM would only support loading all data into main memory upfront. This option can now be accessed in ChromHMM through the '-lowmem' flag. To reduce the time required we used 12-parallel processors ('-p 12' flag) and we trained on a different random subset of the human genome on each iteration of the Baum-Welch algorithm. We divided each chromosome into 200 kb segments (with the exception of the last segment of each chromosome which was less than this) in order to form random subsets of the human genome. We modified ChromHMM to allow training for each iteration on a randomly selected subset of 150 of these segments ('-n 150' flag), corresponding to 30MB per iteration. We ran this for 200 iterations by adding the '-d -1' flag, which removed one of ChromHMM's default stopping criterion based on computed likelihood change on the sampled data, since the likelihood is now expected to both increase and decrease between iterations as different segments are sampled. These new options were included in version 1.13 of ChromHMM. The unique code to ConsHMM v1.0 is written in Python. The code of ConsHMM shared with ChromHMM is written in Java and included with ConsHMM.

**Generating genome-wide annotations.** After ConsHMM learned a state model, we used it to segment and annotate the human genome at base-pair resolution into conservation states. Each base in the human genome is classified into the state with the highest posterior probability. ConsHMM does this by running the MakeSegmentation command of ChromHMM. Due to computational constraints, the segmentation could not be generated for entire chromosomes at once. Instead, we ran MakeSegmentation on the same 200 kb partitioning made for learning the model. We then merged the resulting files together using ConsHMM's mergeSegmentation.py command with slice size parameter set to 200,000 ('-s 200000' flag) and the number of states parameter set to 100 ('-n 100' flag).

**Computing enrichments for external annotations.** All overlap enrichments for external annotations were computed using the ChromHMM OverlapEnrichment command at single base resolution ('-b 1' flag). OverlapEnrichment computes enrichments for an external annotation in each state assuming a uniform background distribution. Specifically, the fold enrichment of a state for an external annotation is

$$\frac{\text{\% of external annotation bases falling in that state}}{\text{\% of genome falling in that state}}$$

Positional enrichments of states relative to an anchor point from an external annotation were computed using the ChromHMM NeighborhoodEnrichment command at single base resolution ('-b 1' flag), single base spacing from the anchor point ('-s 1') and using the '-l' and '-r' flags to specify the size of the region of interest around the anchor point. The '-lowmem' flag was also used for computing the enrichments for OverlapEnrichment and NeighborhoodEnrichment.

**External data sources for enrichment analyses.** The external annotations of repeat elements were obtained from the UCSC genome browser RepeatMasker track[21,39]. We generated an annotation for whether a base overlapped any repeat element, as well as separate annotations for bases falling in each class and family of repeat elements. The gene annotations were obtained from GENCODE v19 for

hg19[51]. CpG island annotations were obtained from the UCSC genome browser. Annotations of SNPs with >=1% minor allele frequency were obtained from the Common SNPs (147) track from the UCSC genome browser, which is based on dbSNP build 147. GWAS catalog variants were obtained from the NHGRI-EBI Catalog, accessed on 5 Dec 2016[48]. For annotations of DNase I Hypersensitive Sites (DHS) processed by the Roadmap Epigenomics Consortium, we used Macs2 narrowPeak calls[5]. The Fetal Brain and HepG2 DHS used were of epigenome samples E082 and E118 respectively. For the median non-exonic DHS enrichments and ranking of states in the heritability partitioning analysis we used narrowPeak calls from the ENCODE consortium[3]. In the cases where ENCODE provided more than one replicate for a cell or tissue type, we used the first replicate.

PhyloP and PhastCons scores and constrained element calls were obtained from the UCSC genome browser. Assembly gap annotations were obtained from the Gap track from the UCSC genome browser. The context-dependent tolerance score (CDTS) used was that based on a cohort of 7784 unrelated individuals, following the analyses in ref. [47], which focused on this version of the score. The CDTS and variants from this cohort were both lifted from hg38 to hg19 using the liftOver tool from the UCSC genome browser[21].

**Choice of number of states**. We learned models with each number of states between 2 and 100 states. We set 100 as the maximum number of states we would consider for computational tractability and maintaining a manageable number of states for analysis. The choice of a maximum of 100 also corresponds to the number of species used and allows for the possibility of each state to cover 1% of the genome. We analyzed the Bayesian Information Criterion (BIC) for models with each number of states between 2 and 100, and found that the BIC generally decreases as the number of states increases in the range considered (Supplementary Fig. 1). The BIC was calculated using the BIC_HMM function from the HMMpa R package[52]. Analyzing the 100-state model's internal confidence estimate of its state assignments also supported a larger number of states. Specifically, for each state in the 100-state model we computed the average posterior probability of that state at each base in the genome assigned to it, and confirmed consistently high average posterior probability values in the range [0.92, 1.00] with a median of 0.97 (Supplementary Fig. 2). The posterior probabilities were computed by running the MakeSegmentation command in ChromHMM with the '-printposterior' flag. We also investigated if additional states in models with larger number of states were biologically relevant. Specifically, we computed enrichments for various external annotations for models with each number of states between 2 and 100 to determine if biologically relevant enrichments were only robustly observed in models with more than a certain number of states. In the case of CpG islands, we observed that only models with at least 87 states consistently obtained >15 fold enrichment and only models with at least 95 states consistently obtained >30 fold enrichment (Supplementary Fig. 3). We saw a similar pattern of increasing enrichments for annotated TSS for models with large number of states. We therefore decided to analyze the largest model, 100 states, that we were considering. We note that annotations based on chromatin states used fewer number of states, but were also defined on fewer features at a coarser resolution and had a less uniform genome coverage[4,30,40].

**State clustering**. We clustered the states based on the correlation of vectors containing the values $f_{k,j}$ and $f_{k,j} \times g_{k,j}$ for each species $j$ defined above. State clustering was performed using the hclust hierarchical clustering function from the cba R package[53]. The leaves of the resulting hierarchical tree were ordered according to the optimal leaf ordering algorithm[54] implemented in the order.optimal R function from the cba package. We then cut the tree such that the 8 major groups of states were designated. The full tree is provided in Supplementary Fig. 4.

**Genome segmentation using uniform transition probabilities**. For analyzing the effect of the transition probabilities on the genome segmentation, we created a separate model, which was the same model we used in the main analyses, except we set all transition probabilities to 0.01, corresponding to each state having an equal probability of transitioning to any state including itself. We then created a new genome segmentation by running the MakeSegmentation command in ChromHMM with this new model. For each state, we counted how many of the bases assigned to it in the original annotation were also assigned to it in the annotation created with the uniform transitions, and divided this number by the number of bases in the state in the original annotation. This calculation provided a fraction from 0 to 1. We also reported the number of segments produced by each model, where a segment is defined to be one or more consecutive bases all assigned to the same state, such that any immediately adjacent bases are assigned to a different state or states.

**GO enrichments**. For each state and each protein-coding gene based on GEN-CODE, we computed the number of bases in that state that are within $+/-2$ kb of the gene's TSS. In the case of genes with multiple annotated TSS, we used the outermost TSS. We then created a ranking of genes for every state by sorting the genes in descending order of this number of bases. For each state, we then created a set of 969 genes that represent the top 5% of genes in the state among the 19,397 genes we considered. We performed a GO enrichment analysis (ontology and

annotations files from 24 November 2016) for the top 5% genes in each state using the STEM v1.3.10 software in batch mode with default options and the set of all genes considered as background[55]. STEM computed an uncorrected p-value based on the hypergeometric distribution for each term displayed in the figures summarizing the analysis. STEM also reported corrected p-values for testing multiple GO terms for a single state based on randomization to three significant digits, which was less than 0.001 for all p-values mentioned in the main text.

**Transcription factor-binding site motif enrichments**. We computed the fold enrichment of the conservation states within 15 bases upstream and downstream of the center point of the *POU5F1* and *STAT* known transcription factor-binding site motifs[37]. The enrichment was computed relative to the background regions of the genome that were used to identify the motifs, which excluded repeat elements, coding sequence, and 3′ untranslated regions (UTRs). We used the *known1* version of the motifs for both *POU5F1* and *STAT*.

**Clustering of cell-type specific DHS enrichments**. For the clustering of DHS analysis, we first computed the fold enrichments of all conservation states for DHS for 53 samples processed by the Roadmap Epigenomics consortium[5], of which 14 were originally generated by the ENCODE project consortium[3]. We then selected the subset of states that had a fold enrichment of at least two in at least one sample, leading to a subset of 21 conservation states. To more directly focus on each state's relative enrichments across samples, we log₂ transformed each enrichment value, and then normalized the enrichments for each state by subtracting the mean enrichment across samples and dividing by the standard deviation. We then hierarchically clustered the states based on the correlation of their enrichments across samples and hierarchically clustered the samples based on their correlations across states using the pheatmap R package[56]. We also computed for each sample the fold enrichment of DHS bases for bases in CpG islands, as the ratio between the percent of DHS bases in CpG islands and the percent of the genome falling in CpG islands.

**Precision recall analysis for recovery of gene annotations**. We randomly split the 200 kb genome segments used for training the model and segmentation into two halves corresponding to training and testing data. For each target set in the precision-recall analyses, we ordered the ConsHMM states in decreasing order of their enrichment for the target among the training set bases. We then used that ordering to iteratively add the testing set bases in each state to form cumulative sets of bases predicted to be of the target set, and computed the precision and recall for them. For each constraint score, we computed the precision-recall curve for predicting the target set in the test data using two methods. For the first method, we directly ordered bases in descending order of their assigned score. For the second method, we split the sorted scores into 400 bins such that each bin contains on average 0.25% of the genome, which was the size of the smallest state of the ConsHMM model (0.25% of the genome in state 100). Specifically, we assigned all bases in the genome where the score was not defined to one bin and then divided the remaining bases uniformly among the 399 other bins based on their score. In some cases, score increments were at the boundary between two bins at their provided floating-point precision, or overlapped multiple bins. In these cases, we uniformly split the target bases assigned to that score increment into multiple bins proportionally to the overall percentage of the score increment falling in each bin. We then treated the 400 bins as 400 states and followed the same procedure described for the ConsHMM states. We also computed the precision and recall of bases in each constrained element set for predicting the target set on the testing data.

**Precision recall analysis for recovery of DHS**. For the precision recall analysis for recovery of DHS analysis for a single cell type, we followed the same procedure described above. We also separately evaluated recovery of DHS bases when restricting the analysis to non-exonic regions. Additionally, both genome-wide and within non-exonic regions, we evaluated the recovery of DHS bases when restricting the analysis to bases distal to a TSS, defined as more than 2 kb from a TSS. For the analysis of the recovery of DHS aggregated across cell and tissue types we concatenated DHS from 53 cell or tissue types processed by the Roadmap Epigenomics Consortium into one annotation in which each combination of chromosome and cell or tissue type effectively becomes a new chromosome. We then split the concatenated data into training and testing sets as described above. We computed the enrichments of the ConsHMM states and scores split into bins as detailed above, but multiplying the size of each state and bin by the number of DNase I hypersensitivity data sets. The precision and recall values for the ConsHMM states, constraint scores considered directly, constraint scores split into bins, and constrained element sets were then computed on the testing data.

**Enrichment analysis for phylogenetically partitioned CNEEs**. We lifted over the CNEEs from ref. [22] from hg18 coordinates to hg19, using the liftOver tool from the UCSC genome browser with default settings[21]. These elements were previously partitioned into subsets based on the inferred branch point of origin in a phylogenetic tree[22]. We computed the enrichments of the conservation states for all the CNEEs and for each subset of the CNEEs separately, using the OverlapEnrichment

command from ChromHMM at single nucleotide resolution ('-b 1' flag) and using the low memory option ('-lowmem'). We also computed analogous enrichments for CNEEs overlapping PhastCons elements called on the same 100-way alignment that the conservation states were annotated based on. To compute the enrichments of CNEEs for bases in CpG islands we created an annotation consisting of a state for each CNEE subset and one additional state for bases not assigned to any CNEE. We then ran the same OverlapEnrichment command as above to compute enrichments of CNEE bases for non-exonic CpG islands, and non-exonic bases in general. The reported enrichment of CpG islands is the ratio of these two enrichments, effectively computing an enrichment relative to the non-exonic background. The set of non-exonic bases for the enrichment analysis was generated by excluding all bases annotated as an exon in GENCODE v19.

**Heritability partitioning analysis**. The heritability partitioning was performed using the LD-score regression ldsc software[15]. We partitioned the PhastCons constrained elements into two halves based on a ranking of the conservation states. We focused on the PhastCons constrained elements for this analysis, since it was the only element set defined on the same alignments as the conservation states. We focused on halves since the LD-score regression estimates can be unstable for annotations covering too small of a percentage of the genome[15]. To determine the two halves we ranked the conservation states in descending order of median fold-enrichment of non-exonic bases for DHS from 123 experiments from the University of Washington ENCODE group[3]. We then divided bases in PhastCons elements between the top 7 ranked states (1–5, 8 and 28), which contain 51.9% of bases in PhastCons elements, and the bottom 93 states, which contain the other 48.1% of bases in PhastCons elements. We applied ldsc to these two sets for 8 traits (age at menarche, body mass index (BMI), coronary artery disease, educational attainment, height, low-density lipoprotein (LDL) levels, schizophrenia and smoking behavior), all of which were previously considered in a heritability partitioning analysis[15]. We followed the procedure for partitioning heritability as done in ref. [15], including using the baseline annotation set and 500 base-pair windows around annotations to dampen the artificial inflation of heritability in neighboring regions caused by linkage disequilibrium. The baseline annotation set contains a range of annotations including DHS. For our analysis, we first removed the constrained element set already included in the baseline annotation set, then added our two halves of PhastCons elements and finally ran the ldsc software on the full set of annotations.

**Enrichment analysis for variant prioritization scores**. For each variant prioritization score included in the conservation state enrichment analysis of prioritized bases, we extracted the top 1, 5, and 10% of all the bases ranked by each score, both genome-wide and just in non-coding regions. The non-coding regions were defined as the intersection of where the LINSIGHT and FunSeq2 scores provided a value, as these two scores were only defined on non-coding regions. This intersection results in a set of bases covering 90% of the genome that excludes coding regions in addition to other regions filtered for technical reasons by either of the two methods[19,41]. For each score we chose the score threshold that gave us a size for the top set that was as close as possible to the target percentage, which did not always exactly match the target percentage due to the precision of the scores. If a score did not provide a value for a particular base being considered, then that base was assigned to the lowest value of that score, but would still be counted when establishing the percentage thresholds. For the scores that provided separate score values for alternate alleles at a certain position, we used the maximum of the values for all alleles. The state enrichments were then computed using the OverlapEnrichment command from ChromHMM at single base resolution ('-b 1' flag) and with the low memory option ('-lowmem' flag). For the analysis restricted to non-coding regions, we also computed the enrichment of the states for this background region using the same command. The enrichment for each score in a state was then divided by the enrichment of the background region for the state. For the Eigen and Eigen-PC scores we used version 1.1, for FunSeq2 we used version 2.1.6, and for CADD we used both v1.0 and v1.4.

**INSIGHT analysis**. The INSIGHT[49] package was used with parameters of 15% allele frequency threshold, 100 minimum neutral flanking sites and the optimizer method BFGS_DIRECT for the OPT_METHOD flag.

**Reporting summary**. Further information on research design is available in the Nature Research Reporting Summary linked to this article.

## Data availability
The ConsHMM conservation state annotations of hg19 are available at https://doi.org/10.6084/m9.figshare.8162036.v1 and https://github.com/ernstlab/ConsHMM. Data behind Supplementary Figs. 2, 5, 6, 8, 9, 13–16, 23–27 is available in Supplementary Data 2 and additional data behind the main figures and Supplementary Fig. 11 is available in Supplementary Data 3. The input multiple species alignment for producing the conservation state annotations is available at http://hgdownload.soe.ucsc.edu/goldenPath/hg19/multiz100way/. The following URLs contain data sets that were used in the downstream analyses: 25-state chromatin state annotations: http://compbio.mit.edu/roadmap; CADD

score v1.0: http://krishna.gs.washington.edu/download/CADD/v1.0/whole_genome_SNVs.tsv.gz; CADD score v1.4: http://krishna.gs.washington.edu/download/CADD/v1.4/GRCh37/whole_genome_SNVs.tsv.gz; CDTS score: http://www.hli-opendata.com/noncoding/coord_CDTS_percentile_N7794unrelated.txt.gz, http://www.hli-opendata.com/noncoding/SNVusedForCDTScomputation_N7794unrelated_allelicFrequency0.001truncated.txt.gz; CNEEs from ref. [22]: http://www.stanford.edu/~lowec/data/threePeriods/hg19cnee.bed.gz; DANN score: https://cbcl.ics.uci.edu/public_data/DANN/data/; EIGEN and Eigen-PC score: https://xioniti01.u.hpc.mssm.edu/v1.1/; ENCODE DHS: http://hgdownload.cse.ucsc.edu/goldenPath/hg19/encodeDCC/wgEncodeUwDnase/; FATHMM-XF score: http://fathmm.biocompute.org.uk/fathmm-xf/; FIRE score: https://sites.google.com/site/fireregulatoryvariation/; fitCons score: http://compgen.cshl.edu/fitCons/0downloads/tracks/i6/scores/; FunSeq2 score: http://org.gersteinlab.funseq.s3-website-us-east-1.amazonaws.com/funseq2.1.2/hg19_NCscore_funseq216.tsv.bgz; GENCODE v19: https://www.gencodegenes.org/releases/19.html; GERP++ scores and constrained element calls: http://mendel.stanford.edu/SidowLab/downloads/gerp/; GWAS catalog variants: https://www.ebi.ac.uk/gwas/; LINSIGHT score: http://compgen.cshl.edu/~yihuang/tracks/LINSIGHT.bw; Motif instances and background: http://compbio.mit.edu/encode-motifs/; REMM score: https://zenodo.org/record/1197579/files/ReMM.v0.3.1.tsv.gz; Roadmap Epigenomics DHS: http://egg2.wustl.edu/roadmap/data/byFileType/peaks/consolidated/narrowPeak/; SiPhy-omega and SiPhy-pi constrained element calls (hg19 liftOver): https://www.broadinstitute.org/mammals-models/29-mammals-project-supplementary-info.

## Code availability
The ConsHMM software is available through https://github.com/ernstlab/ConsHMM. The ChromHMM software used for enrichment analyses and on top of which ConsHMM is built is available at http://www.biolchem.ucla.edu/labs/ernst/ChromHMM/. The STEM software used for GO enrichment analysis is available at http://sb.cs.cmu.edu/stem/. The ldsc software used for the heritability partitioning analysis is available at https://github.com/bulik/ldsc. The INSIGHT software used for selection analyses is available at http://compgen.cshl.edu/INSIGHT/downloads/INSIGHTpackage/.

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

## Acknowledgements

We thank Ewan Birney and members of the Ernst lab for useful discussions. We thank Ilan Gronau and Ritika Ramani for assistance with running INSIGHT. We acknowledge funding from the US National Institutes of Health (DP1DA044371, R01ES024995, U01HG007912, U01MH105578 to J.E., T32CA201160 to A.A.); US National Science Foundation (CAREER Award #1254200 to J.E.); Kure It cancer research (Kure-IT award to J.E.); Alfred P. Sloan Foundation (Alfred P. Sloan Fellowship to J.E.).

## Author contributions

A.A. and J.E. developed the method, analyzed the results, and wrote the paper.

## Additional information

**Competing interests:** The authors declare no competing interests.

