## [Peer Review File · Communications Biology]

Editorial Note: This manuscript has been previously reviewed at another journal. This document only contains reviewer comments and rebuttal letters for versions considered at *Communications Biology* .

Reviewers' comments:

Reviewer #1 (Remarks to the Author):

First, I would like to declare that the original reviewers are not available to assess the revised manuscript. The following comments reflect my understanding of the revised manuscript as a new reviewer and may not represent the original reviewers' opinions.

In this manuscript, the authors present ConsHMM, a new computational method for analyzing comparative genomic data. Unlike classic phylogenetic models, ConsHMM ignores phylogeny and, instead, clusters nucleotide sites into 100 homogenous conservation states. I agree with the other reviewers that ConsHMM is somewhat less interpretable than classic phylogenetic models, which might be a limitation of the current implementation of ConsHMM. However, I feel ConsHMM still has its own merits and could potentially provide a richer summarization of comparative genomic data.

In particular, it is promising to see that LD-score regression suggests that ConsHMM states may provide orthogonal information on heritability after controlling commonly used genomic features, including classic conservation scores and DNase peaks. Therefore, classic conservation scores ignored some information in whole-genome alignments which can potentially be captured by ConsHMM. ConsHMM states may be valuable features in the future development of variant prioritization tools even though ConsHMM states are less interpretable than classic conservation scores.

I have a few additional comments as follows.

Major comment:

The authors have demonstrated that a subset of ConsHMM states are enriched with genetic variants associated with complex traits after controlling well known genomic features. An equally interesting question is whether ConsHMM states can provide additional information for prioritizing genetic variants associated with severe genetic disorders. This question can be addressed using a supervised framework. For example, the authors may use existing training data from GWAVA or alternative tools and see whether adding ConsHMM states can significantly improve the prediction of known disease variants.

Also, it is interesting to check whether different ConsHMM states are under distinct selection pressures in human populations. For example, I could imagine to stratify conserved bases in phastCons elements by ConsHMM states and then use the INSIGHT model (Ilan Gronau et al., MBE, 2013) to evaluate the strength of natural selection on the distinct ConsHMM groups. If different ConsHMM groups are under distinct selection pressures, ConsHMM may capture additional information on natural selection which is ignored by classic conservation scores.

I feel the section of "Bases prioritized by different variant prioritization scores have distinct conservation state enrichment patterns" is difficult to follow and somewhat descriptive. In particular, different machine learning models are trained on distinct datasets and very often give different predictions, so it is not surprising that they are enriched with different ConsHMM states. Also, I don't feel that ConsHMM provides a lot of new insights into the differences among variant prioritization

methods because ConsHMM states themselves are also hard to interpret. Therefore, I suggest to not make as strong of a statement about the interpretation of variant prioritization scores using ConsHMM.

Reviewer's Comments:

Reviewer #1 (Remarks to the Author):

First, I would like to declare that the original reviewers are not available to assess the revised manuscript. The following comments reflect my understanding of the revised manuscript as a new reviewer and may not represent the original reviewers' opinions.

In this manuscript, the authors present ConsHMM, a new computational method for analyzing comparative genomic data. Unlike classic phylogenetic models, ConsHMM ignores phylogeny and, instead, clusters nucleotide sites into 100 homogenous conservation states. I agree with the other reviewers that ConsHMM is somewhat less interpretable than classic phylogenetic models, which might be a limitation of the current implementation of ConsHMM. However, I feel ConsHMM still has its own merits and could potentially provide a richer summarization of comparative genomic data.

In particular, it is promising to see that LD-score regression suggests that ConsHMM states may provide orthogonal information on heritability after controlling commonly used genomic features, including classic conservation scores and DNase peaks. Therefore, classic conservation scores ignored some information in whole-genome alignments which can potentially be captured by ConsHMM. ConsHMM states may be valuable features in the future development of variant prioritization tools even though ConsHMM states are less interpretable than classic conservation scores.

We thank the reviewer for recognizing the merits of ConsHMM and for appreciating that it can provide a richer summary of the information contained in multiple sequence alignments. We also thank the reviewer for their suggestions, which we have used to strengthen our manuscript. We discuss our response to the specific points below.

I have a few additional comments as follows.

Major comment:

The authors have demonstrated that a subset of ConsHMM states are enriched with genetic variants associated with complex traits after controlling well known genomic features. An equally interesting question is whether ConsHMM states can provide additional information for prioritizing genetic variants associated with severe genetic disorders. This question can be addressed using a supervised framework. For example, the authors may use existing training data from GWAVA or alternative tools and see whether adding ConsHMM states can significantly improve the prediction of known disease variants.

We thank the reviewer for appreciating the analysis of partitioning heritability based on conservation states and raising this question.

We followed the reviewer's suggestion and tried incorporating the ConsHMM state annotations into the GWAVA framework. However, we did not see a significant change in the area under the ROC curve reported in the original paper. We note that this result could occur even if the ConsHMM annotations are useful for variant prioritization. This is the case since GWAVA score is trained on a limited and biased set of curated non-coding variants and is thus not sufficiently powered to retain the rich information in the ConsHMM annotations. The limitations of using curated non-coding variants for variant prioritization scores have been previously recognized, and have motivated supervised approaches based on variant constraint within human and closely related primates (e.g. CADD, fitCons, LINSIGHT). As we demonstrate with the added INSIGHT analysis below, which is the basis of the fitCons and LINSIGHT scores, such approaches would have limitations when applied with the conservation state annotations. The reason for this is the relatively direct overlap of information in the ConsHMM annotations and the supervised information that is the basis of these variant prioritization methods in terms of variation within human and close primates.

That being said, we still expect the ConsHMM annotations can be useful for improving the prioritization of genetic variants. However, effectively leveraging the rich annotations in the ConsHMM annotations will require additional methodological advances on approaches for variant prioritization. We have revised the last paragraph of the discussion to include this point. A new supervised framework for variant prioritization that can effectively leverage the information in the conservation states is something towards which we are actively working, but we feel that its development would be outside the scope of the current manuscript, which focuses on an unsupervised framework for annotating the genome.

Also, it is interesting to check whether different ConsHMM states are under distinct selection pressures in human populations. For example, I could image to stratify conserved bases in phastCons elements by ConsHMM states and then use the INSIGHT model (Ilan Gronau et al., MBE, 2013) to evaluate the strength of natural selection on the distinct ConsHMM groups. If different ConsHMM groups are under distinct selection pressures, ConsHMM may capture additional information on natural selection which is ignored by classic conservation scores.

We thank the reviewer for the suggestion. We applied the INSIGHT model to quantify selection pressures in human population in PhastCons elements stratified by ConsHMM state, bases not in PhastCons elements stratified by ConsHMM state, as well as for all bases in each state. The estimated fraction of bases under constraint in each category and estimated number of divergence

events due to positive selection per kbp as computed by INSIGHT are in the new **Figure S29** and included below.

Figure S29: Results of running the INSIGHT model. (A-C) The estimated fraction of bases under selection as reported by the INSIGHT method within (A) each conservation state, (B) each conservation state after removing bases in PhastCons elements and (C) each state restricted only to bases in PhastCons elements. (D-F) The estimated number of divergence events driven by positive selection per kilobase-pair as estimated by the INSIGHT method within (D) each conservation state, (E) each state after removing only to bases in PhastCons elements and (F) each state restricted only to bases in PhastCons elements. States are colored according to their group as indicated on the right.

When considering PhastCons elements stratified by conservation state, INSIGHT estimates a majority of ConsHMM states, 78%, were estimated to have over 75% of bases under constraint, while 12% of states had less than 50% of bases under selection. We observed similar results when considering bases not in PhastCons elements stratified by ConsHMM states as well as when considering all bases in each ConsHMM state. However, it is important to note that it is unlikely that such a high percentage of the genome is actually under constraint in human populations, as these results would imply. Rather, these results are likely a result of the relatively direct relationship between human variation information contained within the conservation states and the INSIGHT's use of such

information to quantify selection. In terms of INSIGHT's estimate of density of positive selection events, the states that had the greatest density of those events (states 55-57 and 87-89) were also the same states we previously highlighted as having the greatest enrichment of common human variation. We discuss these results in a new paragraph at the end of the results section on '*Enrichment of conservation states for human genetic variation.*'

I feel the section of "Bases prioritized by different variant prioritization scores have distinct conservation state enrichment patterns" is difficult to follow and somewhat descriptive. In particular, different machine learning models are trained on distinct datasets and very often give different predictions, so it is not surprising that they are enriched with different ConsHMM states. Also, I don't feel that ConsHMM provides a lot of new insights into the differences among variant prioritization methods because ConsHMM states themselves are also hard to interpret. Therefore, I suggest to not make as strong of a statement about the interpretation of variant prioritization scores using ConsHMM.

We thank the reviewer for the feedback. In response to this comment, we revised **Figure 6** to make it easier to interpret the states being highlighted. As shown below, we provide in the revised figure a summary of align and match probabilities and a summary of notable enrichments for the highlighted states. As the revised figure shows, most of the specific states prioritized by different variant prioritization scores do have distinct enrichments and interpretations that are discussed in more depth throughout the paper. We have also revised and shortened the text so that it is clearer and more focused in the now renamed results section "*Conservation states highlight different classes of putatively important nucleotides being selectively captured by variant prioritization scores.*"

We emphasize that our results go beyond saying there are differences in the variant prioritization score. Importantly, we show that specific prioritization scores do not prioritize bases in specific conservation states that have both biological enrichments and conservation patterns suggestive of being important. For example, we discuss this to be the case for several scores in the context of State 2, which is the most enhancer enriched state, and State 28, the most promoter enriched state, both of which show some high alignment probabilities with non-mammalian vertebrates. We expect these results will be surprising to many users of those scores.

We note that even in cases when different methods use the same data for training one can see very different state enrichments, as we discuss in the text when comparing the states prioritized with CADD versus DANN. This highlights the difficulty in knowing the types of bases a variant prioritization score would rank highly, even when knowing the data used for its training. This supports the utility of using the conservation state annotations, which capture biologically diverse classes of nucleotides at single nucleotide resolution systematically and

in an unbiased way, to understand the types of bases prioritized by a variant prioritization method. We believe that the revisions we have made to **Figure 6** will make the biological interpretation of those classes of nucleotides more easily accessible. The revised **Figure 6** legend also highlights the presence in the supplement **Table S3**, which contains expanded information and includes all states.

Figure 6A: Fold enrichments of bases ranked in the top 1% of the non-coding genome by 14 variant prioritization scores. Only states that were among the top five most enriched states for at least one score are shown. The enrichment of the top five ranking states for each score is colored according to the ranking and the color scale shown on the left. A summary of the align and match probabilities and notable enrichments of each state is in the table on the right. The 'Distal align' and 'Distal match' columns contain the species most distal to human that has an alignment and matching probability in the state greater than 0.5, respectively. The 'Proximal not align' and 'Proximal not match' columns contain the species closest to human that has an alignment and matching probability in the state lower than 0.5, respectively. The species are colored by the major clades indicated below. An expanded version including all states is available in **Table S3**.

REVIEWERS' COMMENTS:

Reviewer #1 (Remarks to the Author):

All my comments have been properly addressed in the revised manuscript. I recommend this work for publication in Communications Biology.

Reviewer Comments:

Reviewer #1 (Remarks to the Author):

All my comments have been properly addressed in the revised manuscript. I recommend this work for publication in Communications Biology.

Author Response:

We thank the reviewer for the positive comment and recommending publication in Communications Biology.